# Microbial Toxins in Insect and Nematode Pest Biocontrol

**DOI:** 10.3390/ijms22147657

**Published:** 2021-07-17

**Authors:** Subbaiah Chalivendra

**Affiliations:** LSU AgCenter, Baton Rouge, LA 70803, USA; schalivendra@agcenter.lsu.edu or Subbaiahchalivendra@gmail.com

**Keywords:** pore-forming toxins, insect ion channel modulators, innate immunity busters, cyclic lipopeptide surfactants, psychoactive compounds, ribotoxins, sterol homeostasis disruptors, uncouplers

## Abstract

Invertebrate pests, such as insects and nematodes, not only cause or transmit human and livestock diseases but also impose serious crop losses by direct injury as well as vectoring pathogenic microbes. The damage is global but greater in developing countries, where human health and food security are more at risk. Although synthetic pesticides have been in use, biological control measures offer advantages via their biodegradability, environmental safety and precise targeting. This is amply demonstrated by the successful and widespread use of *Bacillus*
*thuringiensis* to control mosquitos and many plant pests, the latter by the transgenic expression of insecticidal proteins from *B*. *thuringiensis* in crop plants. Here, I discuss the prospects of using bacterial and fungal toxins for pest control, including the molecular basis of their biocidal activity.

## 1. Introduction

Insects constitute the largest and most diverse group of animals (~2.5 million species) on Earth [1]. With the exception of a few beneficial insects such as pollinators, they form the costliest animal group to human society by spreading devastating infectious diseases in humans, livestock and crops, ravaging food stocks, damaging forests, destroying infrastructure and weakening the resilience of ecosystems [1]. The Nematoda (also called Nemathelminthes), with an estimated 500,000 species, is the second largest phylum in the animal kingdom [2]. Most species of nematodes are either innocuous or play beneficial ecological and agronomic roles, by nutrient recycling and controlling insects and other harmful nematodes [3]. At the same time, some nematodes cause serious diseases in plants, humans and other animals. For example, *Wuchereria bancrofti*, the most prevalent filarial nematode, affects over 100 million people throughout tropical parts of the world [4]. Plant parasitic nematodes, also called as “eelworms”, are estimated to cause 77 billion dollars of damage worldwide each year [5]. Crop damage due to diseases and pests is expected to worsen due to range expansions caused by climate change and biological invasions, particularly in food security “hotspots” [1,6]. Biocontrol is the means of controlling pests and pathogens through the use of other organisms, which can be natural enemies, such as predators, parasitoids, pathogens and competitors. It is an environmentally safe, low-cost and effective approach and occurs in natural communities. Much before the introduction of chemical pesticides, biocontrol has been in practice to control agricultural pests, with the first recorded report in 304 AD from China [7]. In spite of clear advantages in terms of environment and food safety, microbials have not replaced chemical pesticides or become a major component of integrated pest management in intensive agriculture or the management of human and livestock health, the only exception being the use of *Bacillus thuringiensis* in mosquito control. Reasons for the limited success of biocontrol agents (BCAs) in pest control may include poor adaptation of a BCA to a new host, extensive non-target effects of the biocontrol microbe [8], lack of methods to mass produce fastidious BCAs in synthetic media, e.g., *Paenibacillus popilliae* [9], and environmental risks associated with the BCA, e.g., vancomycin resistance in *P. papillae* [10]. These concerns, exacerbated by the lack of efforts to clearly validate the potential environmental risks, have slowed down the use of microbials in pest control [11]. Many of these constraints can be lessened when the identity of a BCA’s biocidal activity and its mode of action are known, as evident by the enormous success of *Bacillus thuringiensis* crops (*Bt*-crops: crops that are genetically modified to express *crystal* (Cry) proteins of *B. thuringiensis*) [12]; (Figure 1).

Detailed analysis of microbial toxins not only helps in strain improvement of candidate BCAs but also would enable the assessment of their non-target effects and modify target specificity, mass produce the toxins in heterologous systems, incorporate insecticidal activity in crops, develop synthetic versions with enhanced toxicity and specificity, ensure consistency of insecticidal action when needed and develop formulations for potential deployment as tank mixes. There is a huge untapped potential for product development both in the form of microbials or toxins for environmentally safe and effective pest control. In this review, I attempt to provide a comprehensive and current understanding of major microbial (both bacterial and fungal) toxins with proven insecticidal and nematicidal activities. I have grouped them by their mode of action (MOA) and discussed structure–activity relationships where information is available, with a hope that the information may stimulate new lines of research toward product development.

## 2. Pore-Forming Toxins

Pore-forming toxins (PFTs) are the largest class of proteinaceous bacterial toxins and important virulence factors, such as colicins of *E. coli* and anthacin of anthrax. All PFTs are synthesized as water-soluble proteins but subsequently become membrane bound. They recognize host cells based on specific cell surface receptors, which can be proteins, lipids or sugars. The binding allows a rapid increase in the local concentration of a PFT and its oligomerization, which is followed by insertion and pore formation in the host cell membrane. The changes in the permeability of the membrane due to pore formation depends on the toxin, ranging from the loss of small ions, e.g., K^+^ and Ca^2+^, to macromolecules such as proteins [15]. Based on the secondary structure of their membrane-spanning domains, PFTs belong to two large groups, α-PFTs and β-PFTs, each with three families. Crystal protein toxins (Cry toxins) from the aerobic spore-forming Gram-positive bacterium Bacillus thuringiensis are α-PFTs with the colicin fold, which is important for pore formation. They form crystalline parasporal inclusions in the bacterium and are best known for their insecticidal activity [15]. *B. thuringiensis* (*Bt*, hereafter) was first isolated in Japan in 1902 from dead silkworm (*Bombyx mori*) larvae and soon recognized as an entomopathogenic bacterium. *Bt* was one of the first prokaryotic BCAs used as a commercial insecticide in France in 1938 [16]. The insect- and nematode-specific pathogenicity of Cry toxins is due to their oral toxicity, and these toxins are active against a wide range of insects from Lepidoptera, Coleoptera, Hymenoptera and Diptera, as well as nematodes [17,18]. They are represented by approximately 300 proteins divided into 75 subgroups with high target specificity. The species-specific pathogenicity of Cry proteins is determined by their binding specificity to the surface proteins of the target insect midgut epithelial cells [19]. They undergo a sequential binding process with glycosyl-phosphatidyl-inositol anchored proteins such as alkaline phosphatase (ALP) or aminopeptidase-N (APN) and cadherin-like protein of the insect midgut. The interaction leads to membrane insertion of the toxin, pore formation and cell lysis. The 3-Domain (3D) family group is the largest group of Cry toxins with >50 different groups, and these toxins are used worldwide for insect control [20]. Although the domains are not conserved in their sequence, their tertiary structure (folding) is similar. Domain I, α-helix bundle, is involved in toxin oligomerization, membrane insertion and pore formation. Domains II and III, composed of beta sheets with exposed loop regions, are the specificity determinants of Cry toxins. The toxins are made during sporulation of the bacteria as larger protoxins, some twice as large as the processed protein. Larger as well as smaller protoxins are processed by insect midgut proteases, resulting in a core protease-resistant and biologically active 60 kDa toxin [20]. With some exceptions, the core proteins cluster according to their insect host specificity. Divergence, in particular Domain III and a part of Domain II, seems to have promoted the rapid evolution and elaboration of *Bt* adaptation to diverse insect receptors [21,22]. In addition to Cry toxins, *Bt* strains synthesize β-PFTs called cytolytic (Cyt) toxins, which also form parasporal inclusion bodies (Crystals) at the onset of sporulation. Cyt toxins are inserted into the membrane to form a β-barrel composed of β-sheet hairpins from each monomer [20]. In contrast to Cry toxins, Cyt toxins act mainly against Diptera; they do not bind any insect gut proteins but directly interact with insect membrane lipids and insert into the membrane. *Bt* also makes insecticidal proteins during the vegetative growth phase which do not form crystals but are secreted into the growth medium. These vegetative insecticidal proteins (Vips) show activity against lepidopteran, coleopteran and some homopteran pests [23]. Vip proteins differ from the Cry proteins in that they share no sequence homology or membrane binding sites in the insect gut. There have been >150 Vip proteins belonging to four subfamilies reported so far. Of them, only Vip3 is represented by PFTs and is the largest group with 111 members [24]. Pyramiding Vip3Aa with Cry toxins appears to delay the development of field-evolved resistance against *Bt*-crops that express Cry proteins alone [25].

Genetic engineering technology enabled the introduction of genes encoding Cry and other *Bt*-insecticidal proteins into major crops. The first *Bt*-crop (maize) was approved for cultivation in 1995 in the US and its success has allowed a widespread cultivation of *Bt*-maize and other crops in the US and 18 other countries across the globe. However, there are many insect pests that show no or limited susceptibility to Cry toxins, and those susceptible develop resistance. Thus, there is a need to explore for novel insecticidal proteins as well as engineer variants of existing ones by in vitro protein evolution for enhanced toxicity, specificity and target range.

PFTs are not only made by bacteria but also are increasingly being discovered in eukaryotes [26], including as components of vertebrate immune systems [27,28]. The aegerolysins comprise over 350 small (~15–20 kDa), β-structured proteins found in several bacteria and eukaryotes. They are known to interact with specific membrane lipids and lipid domains and have recently shown to be a new class of PFTs. Recently, Panevska et al. [26] discovered that three aegerolysins from the mushroom *Pleurotus* (ostreolysin A, pleurotolysin A and pleurotolysin B) assemble to form multimeric transmembrane pores. These proteins possess the membrane-attack-complex/perforin domain and specifically recognize ceramide phosphoethanolamine, the main sphingolipid in invertebrate cell membranes. Further, these aegerolysin/PlyB complexes show selective toxicity toward Western corn rootworm larvae and adults as well as Colorado potato beetle larvae and may be developed as potent alternatives to *Bt* toxins [26].

## 3. Insect Ion Channel Modulators

Ion channel modulators have clear advantages as pesticidal targets in that they act quickly and allow the rapid reduction of insect pressure. Five ion channels within the insect nervous system have been the primary targets for the development of small molecule insecticides and also serve as effective targets for biopesticides. These are the γ-aminobutyric acid (GABA) receptor, the glutamate-gated chloride channel, the nicotinic acetylcholine receptor or nAChR, the voltage-gated sodium channel and the ryanodine receptor [29]. So far, microbial toxins that target the glutamate-gated chloride channel (Glu-Cl) and ryanodine receptor (RyR) have been identified. Macrocyclic lactones bind to an allosteric site on Glu-Cl, either directly activating the channel or enhancing the effect of the endogenous agonist, glutamate. Avermectins, with a large macrocyclic lactone ring produced from the metabolites of Gram-positive bacterium *Streptomyces avermectinius*, are well-known examples of Glu-Cl modulators. They act on insects/nematodes preventing the transmission of electrical impulses in the muscles and nerves of invertebrates and are well-known drugs and pesticides. They amplify the glutamate effects on the invertebrate-specific gated chloride channel, activate both ligand- and voltage-gated chloride channels and induce paralysis in insects [30,31]. Nodulisporic acid A, a natural insecticidal indole terpene isolated from an endophytic fungus, *Nodulisporium* sp., also inhibits only invertebrate-specific glutamate-gated chloride (Glu-Cl) channels [32]. Based on its safety, Merck & Co. introduced synthetic versions of the compound as a potent oral formulation to control fleas and ticks in dogs and cats [33]. Ryanodine receptors (RyRs) belong to a group of ligand-gated calcium channels initially reported from vertebrates. They are located on the endoplasmic reticulum of muscle cells and neurons and play a critical role in muscle contraction. In contrast to mammals, which have three types of RyRs, insects have only a single RyR, which is a major target for modern insecticides. Ryanodine, a plant alkaloid and an important ligand of RyR, has served as a natural botanical insecticide. Attempts to generate synthetic commercial analogs of ryanodine have, so far, not succeeded. Despite the popularity of diamide insecticides due to their specificity, several agricultural pests have become resistant to the chemical insecticides due to mutations in a transmembrane region of their RyRs [34], and there is a need for alternative RyR agonists. Cyclodepsipeptides are a large family of peptide-related natural products with an α-hydroxy acid and 5–10 amino acids linked by amide and ester bonds [35]. Verticilide, a cyclooctadepsipeptide produced by *Verticillium* sp. FKI-1033, has been shown to bind selectively to the insect ryanodine receptor in the low micromolar range [36]. The edible oyster mushroom, *Pleurotus ostreatus*, is also known for its nematophagy. It produces powerful toxins that immobilize nematodes within minutes of contact and feeds on dead nematodes. One of these toxins is trans-2-decenoic acid (TDA), a toxin with nematotoxic action derived from linoleic acid. The toxin affects not only nematodes but also insects and even other fungi [37]. Recent work has provided more details on the mechanism of action of the toxin, suggesting that TDA may act as a RyR ligand [38]. Intact sensory cilia on the nematode’s head, hairs that help it sense its surroundings, are required for the TDA toxin to cause paralysis. Lee et al. [38] found that a substantial influx of Ca^2+^ levels occurs in the pharynx and head muscles of nematodes treated with TDA. Genetic evidence suggested that TDA-caused Ca^2+^-influx, neurotoxicity and nematode paralysis are mediated by the activation of the RyR channel.

## 4. Innate Immunity Busters

Insects, like all invertebrates, express both innate and humoral immunity to infection. Insect immunity is usually resolved into three major components: (1) the integument, which serves as a physical barrier to infections, (2) the circulating hemocytes responsible for clearing the majority of infecting bacteria from the hemocoel and (3) specific cellular defenses including phagocytosis, microaggregation of hemocytes with adhering bacteria, nodulation and encapsulation [39]. Infections also stimulate the humoral component of immunity, which involves the induction of antimicrobial peptides and the activation of prophenoloxidase (PO). The PO system is responsible for melanization, i.e., the synthesis and deposition of melanin—the hard black polymer that seals off microbial pathogens [39]. Prostaglandins and other eicosanoids are crucial mediators of innate immune responses whose biosynthesis is mediated by phospholipase A2 (PLA2). The inhibition of PLA2 and thereby eicosanoid biosynthesis is lethal to insects. Biological control strategies that are targeted to these pathways have a great potential for success [40].

Bacteria belonging to 24 species of *Xenorhabdus* and five of *Photorhabdus* are known worldwide for their entomopathogenic potential. Some species (e.g., *P. luminescens* and *X. nematophila*) have mutualistic associations with nematodes and share their entomophagous lifestyle. The soil-dwelling entomopathogenic nematode larvae find and penetrate the insect larvae through natural openings and release symbiotic bacteria into the insect hemocoel. The bacteria replicate in the insect body and release immuno-suppressive toxins. At least seven secondary metabolites that inhibit PLA2, namely, benzylideneacetone (BZA), proline-tyrosine, acetylated phenylalanine-glycine-valine, indole, oxindole, cis-cyclo-PY and p-hydroxyphenyl propionic acid, have been reported. They also show significant inhibitory activities against other immune responses, such as phenoloxidase activity (PO) and hemocytic nodulation, with BZA being the most effective [41]. An isocyanide-containing compound rhabducin produced by *Xenorhabdus* can also inhibit phenoloxidase and thereby melanization [42]. In addition, phurealipids (urea compounds) made by these bacteria can prevent the expression of antimicrobial peptide genes in the insect hosts, a part of the humoral component of immunity [43]. It is no surprise that these bacterial products induce fatal septicemia and toxemia culminating in the death of target insects in 24 to 48 h [44]. They offer promising biocidal candidates for more detailed testing individually or in combination.

## 5. Cyclic Lipopeptide Surfactants

Cyclic lipopeptides (CLPs) are amphiphilic molecules composed of a cyclic oligopeptide lactone ring coupled to a fatty-acid tail. Although they are as short as 7–10 amino acids, CLPs show remarkable heterogeneity. In addition to amino acid substitutions, the nature, length and configuration of the fatty-acid chain contribute to the variability. Their structural diversity and cyclic configuration confer them with broad-spectrum and environmentally stable antibiosis activity against bacteria, fungi, insects, protozoa and even human tumor cell lines [45,46]. CLPs are produced mostly by *Pseudomonas*, *Bacillus* and *Streptomyces* spp. CLPs come in three families, namely surfactin, iturin and fengycin, and a single strain can make one or more of them contribute to varying biocidal activities. *Pseudomonad* spp. (e.g., *P. chlororaphis*, *P. fluorescens*, *P. protegens*, *P. putida*, *P. mosselli* and *P. entomophila*) are particularly pathogenic toward insects and nematodes. Major CLPs produced by Pseudomonas spp. are biosurfactants. In particular, orfamides, consisting of 10 amino acids and a 3-hydroxydodecanoic or tetradecanoic acid tail, show strong biosurfactant activity and anti-aphid insecticidal activity both by oral and external application through the cuticle [47,48,49,50]. The aphicidal activity of *Bacillus* spp. (*B. atrophaeus* L193, *B. subtilis* Y9 and *B. amyloliquefaciens* G1) has also been attributed to lipopeptide biosurfactants that damage the cuticular exoskeleton [51]. Orfamides from Pseudomonas and surfactin CLPs from Bacilli also show strong biocidal activities against fungi including oomycete pathogens, thus making them versatile biopesticidal candidates [52,53,54].

## 6. Psychoactive Compounds

Many microbial parasites manipulate host insect behavior for the enhanced dispersal of their inoculum and thereby spread disease. A few of them, such as baculoviruses, hijack host genes involved in insect physiology [54], whereas others make their own [55]. Many entomopathogenic fungi cause “summit disease” (SD) behavior, an extended phenotype where parasitized insects ascend and affix to elevated substrates prior to death. This facilitates a wider dissemination of fungal spores from the mummified insect carcasses. For example, the fungal pathogen *Entomophthora muscae* infects wild *Drosophila* and manipulates host behavior. In response to *E. muscae* infection, flies climb up a grass blade, attach to it and then lift their wings, leading to the dissemination of infectious spores from their abdomens. Although the molecular basis of this behavior manipulation is not yet clear, the fungus has been shown to invade the nervous system [56]. It is also speculated that *E. muscae* may produce and secrete eicosanoid-like compounds to induce behavioral changes in the dying host [57]. More details are available about the bizarre behavior in cicadas induced by the pathogenic fungi *Massospora* and *Strongwellsea*. *M. cicadina*, *M. platypediae*, *M. levispora*, *S. tigrinae* and *S. acerosa* keep their insect hosts alive while sporulating, which enhances dispersal via sexual transmission. Conidia are actively disseminated from the posterior end of the infected cicada during mating attempts and flights, even after half of their bodies are digested and replaced by fungal biomass. Metabolomics analyses suggest that the fungus modifies the host behavior using neurogenic metabolites such as cathinone, tryptamine and psilocybin, as well as enterotoxins [55]. Species belonging to the “zombie ant fungus” *Ophiocordyceps* (e.g., *O. kimflemingiae*; *O. camponoti-floridani*) are ant-feeding fungi that use enterotoxins and behavior-manipulating chemicals to manipulate their hosts. Infected ants show an increase in activity and wandering behavior followed by a final summiting and biting behavior onto vegetation. This aberrant behavior positions the manipulated ant in a site beneficial for fungal growth and transmission. Fungal-secreted enterotoxins seem to modify ant behavior by interfering with the production of chemosignaling molecules, in addition to killing the host [58,59]. More research is needed to validate the findings from -omics analyses to unequivocally implicate any of these putative chemosignaling compounds in insect behavioral manipulation. Nevertheless, this concept has been successfully exploited in the management of some pests, e.g., codling moth (*Cydia pomonella*, a key insect pest of apple), using host endogenous molecules, especially pheromones [60].

## 7. Ribotoxins

Ribosome-inhibiting proteins (RIPs) or ribotoxins are produced by *Aspergillus* and *Penicillium* spp. and also by some entomopathogenic fungi, e.g., *Hirsutella thompsonii* or *Metarhizium anisopliae* [61]. RNase T1 is the best-known representative of this group of cytotoxins. Ribotoxins can enter the cell by crossing lipid membranes without the need for a protein receptor via their ability to interact with acid phospholipid-containing membranes. Once inside the cell, they cleave a single phosphodiester bond located within the sarcin–ricin loop, a universally conserved sequence of the large rRNA gene. This leads to the inactivation of ribosomes, the inhibition of protein biosynthesis, followed by cellular death by apoptosis [62]. *Hirsutella thompsonii* infects different types of insects as well as mites and nematodes, particularly causing spectacular natural epizootics among mite populations [63]. The fungus produces extracellular insect toxins, one of which is called as Hirsutellin A toxin (HtA). HtA shares partial sequence homology and biological properties with known ribotoxins and may prove to be a promising novel biopesticide and nematicide. There are already efforts to commercialize ribotoxin formulations as baculovirus-based biopesticides [64].

## 8. Sterol Homeostasis Disruptors

Most insects are sterol auxotrophs and depend on dietary sterols for their structural and metabolic needs. Therefore, cholesterol uptake, transport and metabolism can be potent targets for vector and pest control strategies [65]. A secreted cholesterol oxidase from *Streptomyces* shows strong insecticidal activity against boll weevil larvae [66]. Beauveriolide III, a depsipeptide made by *Beauveria* sp., selectively inhibits sterol O-acyltransferase 1 in a cell-based assay. Sterol O-acyltransferase catalyzes the formation of fatty-acid cholesterol esters and is important for dietary cholesterol absorption. Pyripyropenes from the fermentation broth of *Aspergillus fumigatus* (an opportunistic human pathogen that causes invasive pulmonary aspergillosis) are also strong inhibitors of acyl-CoA:cholesterol acyltransferase [67]. In foliar sprays and soil drenches, pyripyropene A has shown high insecticidal activities against some sucking pests, such as whiteflies and, particularly, aphids. The compound also has a low eco-toxicological impact and satisfies a prerequisite for next-generation insecticides, highlighting its commercialization potential [68]. Some derivatives showed a higher inhibitory effect on canine heartworms and mosquito vectors than did the parent compound, suggesting their utility as filaria control drugs [69]. One of the derivatives showed far superior activity to that of the natural compound on aphids and was commercialized as afidopyropen (Figure 2) [70], demonstrating the power of synthetic chemistry tools in product discovery.

## 9. Uncouplers and Electron Transport Inhibitors

Natural compound inhibitors of the mitochondrial electron transport chain and uncouplers of oxidative phosphorylation are promising candidate insecticides if they also prove to be highly selective and environmentally safe. The well-known example is the isoflavone rotenone, a complex I inhibitor from plants. It is a broad-spectrum insecticide and piscicide, due to its mild toxicity to humans and other mammals [71]. A few microbial toxins with insecticidal properties show this mode of action. A common soil hyphomycete, *Purpureocillium lilacinum* (syn. *Paecilomyces lilacinus*), is used as a nematicide. However, it can cause rare opportunistic infections in humans and also has significant resistance to conventional antifungals [72]. It secretes a nematicidal toxin called paecilotoxin. The toxin belongs to the leucinostatins, a class of neutral straight peptides containing an unsaturated fatty acid at the N-terminus, and has strong uncoupling activity via its inhibition of the membrane-bound component of ATP synthases [73]. Overproduction of the toxin in fermentation may allow large-scale field testing for its pesticidal efficacy and environmental safety [74].

## 10. Conclusions

Although microbials are ecologically friendly alternatives to chemical pesticides, their use for pest control (BCAs) has been limited for a number of reasons. Major constraints include cost, adaptation to new environments, low virulence of the organism and the length of time it takes to kill its target, and in a few cases potential non-target effects. Knowledge on the biocidal component of the BCA allows the development of more effective and targeted pest control technologies, as exemplified by the widespread success of *Bt*-crops. Genome-guided approaches and cutting-edge metabolomics tools have greatly empowered our efforts to reveal the bioactive compounds that confer the biocontrol effect. Advances in synthetic chemistry and innovative model systems for high-throughput screening of candidate compounds are further speeding up the discovery and product development of cost-effective, environmentally safe and robust pest control measures.

## Figures and Tables

**Figure 1 ijms-22-07657-f001:**
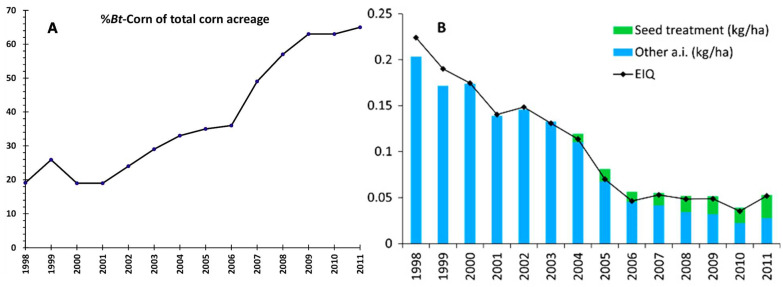
Insect resistant *Bt*-corn decreased pesticide use in US. (**A**) Percentage of *Bt*-corn acreage out of total corn planted between 1998 and 2011 (data extracted from [13]). (**B**) Insecticide use in maize (kg/ha and environmental impact quotient (EIQ) weights) during the same period as in A. (Figure reproduced from [14]).

**Figure 2 ijms-22-07657-f002:**
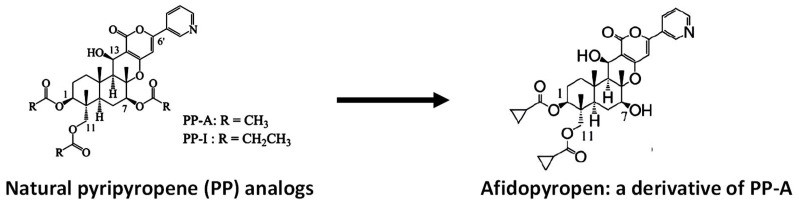
A derivative of natural pyripyropene shows superior aphicidal activity. Of >40 derivatives tested, the derivative with cyclopropanecarbonyloxy groups at the C-1 and C-11 positions and a hydroxyl group at the C-7 position showed the highest insecticidal activity against aphids [70].

## Data Availability

Not applicable.

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
