# Peer review of "Microbial Toxins in Insect and Nematode Pest Biocontrol"

_ijms, 2021, doi:10.3390/ijms22147657_

Round 1

Reviewer 1 Report

The manuscript is a nice overview of the toxins produced by some enotmopathogenic micribials. It is of interest to the readership in the field.

However it is a short representation of a complex field of research, and it should be seen more as a "Communication" providing several examples instead of an inclusive "Review".

Line 194: humoral response should be included in the list of main innate immunity

Author Response

Reviewer: However it is a short representation of a complex field of research, and it should be seen more as a "Communication" providing several examples instead of an inclusive "Review".

Response: I agree with the reviewer that it is a complex field of research. That is why I have taken a novel (different) approach and used mode of action to make the review comprehensive and yet not too long or redundant with the existing literature. There have been many topical reviews on specific biocontrol organisms published already and they have been referred in this manuscript for the benefit of the reader.

Reviewer: Line 194: humoral response should be included in the list of main innate immunity.

Response: I thank the reviewer for pointing out this and I made the suggested change. Although I have listed the humoral response in the subsequent, I missed it in the introductory sentence.

All changes are indicated in red font.

Reviewer 2 Report

Dear Author it is very good ms but add reference in a large amount or paragraphs in the text and the conclusion is not proper. Also check the reference in the txt.

I have made my suggestion in the pdf file

Author Response

Reviewer: Dear Author it is very good ms but add reference in a large amount or paragraphs in the text and the conclusion is not proper. Also check the reference in the txt.

Response: Thank you very much for the kind words and suggestions for improvement. I have added new references now wherever marked and changed the conclusion part. 

Reviewer: I have made my suggestion in the pdf file.

Response: I have incorporated all suggestions as per my understanding of reviewer's comments.

Round 2

Reviewer 2 Report

Dear author i believe is ready to published. Very good review